# Ballistic Coefficient Calculation Based on Optical Angle Measurements of Space Debris

**DOI:** 10.3390/s23187668

**Published:** 2023-09-05

**Authors:** Yigao Ding, Zhenwei Li, Chengzhi Liu, Zhe Kang, Mingguo Sun, Jiannan Sun, Long Chen

**Affiliations:** 1Changchun Observatory, National Astronomical Observatories, Chinese Academy of Sciences, Changchun 130117, China; 2University of Chinese Academy of Sciences, Beijing 100049, China; 3Key Laboratory of Space Object and Debris Observation, Purple Mountain Observatory, Chinese Academy of Sciences, Nanjing 210008, China

**Keywords:** ballistic coefficients, orbit prediction, mean element

## Abstract

Atmospheric drag is an important factor affecting orbit determination and prediction of low-orbit space debris. To obtain accurate ballistic coefficients of space debris, we propose a calculation method based on measured optical angles. Angle measurements of space debris with a perigee height below 1400 km acquired from a photoelectric array were used for orbit determination. Perturbation equations of atmospheric drag were used to calculate the semi-major-axis variation. The ballistic coefficients of space debris were estimated and compared with those published by the North American Aerospace Defense Command in terms of orbit prediction error. The 48 h orbit prediction error of the ballistic coefficients obtained from the proposed method is reduced by 18.65% compared with the published error. Hence, our method seems suitable for calculating space debris ballistic coefficients and supporting related practical applications.

## 1. Introduction

Most space applications and studies have increasingly improved the requirements for orbit calculation accuracy. As typical non-conservative forces, atmospheric drag perturbations have a long-term and significant impact on the orbit of space targets [1,2,3]. For accurately calculating atmospheric drag, the ballistic coefficients of space debris are important factors, and the acquisition of accurate ballistic coefficients allows us to determine and predict the orbits of low-orbit space debris [4].

The ballistic coefficient of a generic body can be expressed as [5]
(1)$$B=C_{D}\frac{A}{m},$$
where $C_{D}$ is the atmospheric drag coefficient of the target, A is the cross-sectional area with respect to the velocity direction of the target relative to the atmosphere at a given time, and m is the mass of the target. The ballistic coefficients of different space objects vary over time. Therefore, the average semi-major-axis change caused by atmospheric drag can be calculated by using the average number of space debris. Then, the ballistic coefficient can be obtained by using the drag perturbation equation of the semi-major axis [5,6,7,8]. This method uses the average number of elements in the orbital data of approximately 17,000 target tow line elements (TLEs) updated daily by the North American Aerospace Defense Command (NORAD) through its space tracking website. The calculation of a space target’s orbit using the TLE orbit report requires the SGP4 model developed by NORAD. The SGP4 model was developed by Ken Cranford in 1970 and is used for near-Earth satellites [9]. This model is a simplification of Lane and Cranford’s (1969) extensive analytical theory, which takes into account the effects of perturbations such as Earth’s non-spherical gravity, solar lunar gravity, solar radiation pressure, and atmospheric drag. SGP4 (Simplified General Perturbations) is a simplified conventional perturbation model that can be applied to near-Earth objects with orbital periods less than 225 min [10,11]. Then, it calculates the atmospheric drag perturbation on the semi-major axis of space debris and estimates the ballistic coefficient. This method provides the estimated ballistic coefficients of multiple low-orbit space targets for quality verification of TLE data. In 2020, Wei et al. [12] proposed an iterative calculation method based on multiple sets of TLEs from public data of space targets to reduce the influence of abnormal TLEs on the calculation results and obtain the ballistic coefficients of space debris. In 2022, Kuai et al. [13] collected a training set to predict ballistic coefficients by using space debris TLEs, a simplified general perturbation model (SGP4), and the publicly available object falling time as measured data samples. They used the iterative correction of ballistic coefficients to construct a long short-term memory neural network to predict the ballistic coefficients.

Considering the previous research, we propose a method for calculating the ballistic coefficients of space debris using optical angle measurements. The data acquisition of this method is more convenient compared to radar measurement data. The basic principle of the method is as follows:(1)First, NORAD TLE space debris data are used to identify the optical angle measurements of massive space debris obtained from the photoelectric telescope array at the Changchun Observatory.(2)Then, the recognition results are combined with corresponding observations to determine the orbit. The orbit determination results are used to infer the mean elements of space debris at a specific time.(3)Finally, the average ballistic coefficient of the corresponding space debris arc segment is calculated using the change in the semi-major axis of the corresponding elements and atmospheric model. To validate the calculations using our method, we used extrapolated ephemeris calculations for comparison.

## 2. Calculation of Ballistic Coefficients from Optical Angle Measurements

### 2.1. Mean Elements of Space Debris

Owing to the inability of space debris telescope arrays to continuously track observation targets, publicly available data of two-line elements should be used for space debris orbit recognition before orbit calculation. Hence, the joint identification and processing of massive unknown target data generated by space debris telescope arrays should be performed. The calculation of the mean elements of space debris based on optical angle measurements is described in Figure 1.

We apply the gradient descent method to calculate the mean elements [14,15,16]. In addition, we use the SGP4 model as follows:(2)$$(y_{1},y_{2},y_{3},y_{4},y_{5},y_{6})=f(x_{1},x_{2},x_{3},x_{4},x_{5},x_{6}),$$
where y is the orbital parameter PosVel, and x represents the orbital elements described by Semimajor axis (a), Eccentricity (e), Inclination (i), Longitude of the ascending node (Ω), Argument of periapsis (ω), and Mean anomaly at epoch (M). In Equation (2), f denotes the SGP4 model comprising dozens of equations. The SGP4 model can provide the orbital parameters at a specific time from the mean elements [17,18,19,20]. The orbit parameter ${PosVel}_{i}$ of the orbit determination result is used to reverse calculate the instantaneous orbital elements $x_{i}$. Then, instantaneous orbital elements $x_{i}$ are reversed to obtain $y_{i}$ for different time instants t. We also calculate the root mean square error (RMSE) as follows:(3)$$RMSE\left(x\right)=\sum_{-t}^{t}\frac{\sqrt{\sum {\left(y_{it}-{PosVel}_{it}\right)}^{2}}}{\left(2t+1\right)},$$

The partial derivative of the RMSE with respect to each orbital element is obtained numerically as follows:(4)$$\frac{\partial RMSE}{\partial x_{i}}=\frac{RMSE(x_{i}+∆x_{i})-RMSE(x_{i}-∆x_{i})}{2∆x_{i}},$$

We perform fitting and calculation using the Adam gradient method. Unlike the conventional gradient descent method, the Adam gradient can automatically adjust the step size at each iteration as follows [21,22,23,24,25]:(5)$$m_{t}=\beta_{1}\cdot m_{t-1}+(1-\beta_{1})\cdot g_{it},$$
(6)$$v_{t}=\beta_{2}\cdot v_{t-1}+\left(1-\beta_{2}\right)\cdot g_{it}^{2},$$
(7)$$\hat{m}=\frac{m_{t}}{1-\beta_{1}^{t}}\;\hat{v}=\frac{v_{t}}{1-\beta_{2}^{t}},$$
(8)$$x_{i_{t}}=x_{i_{t-1}}-\alpha \cdot \frac{\hat{m}}{\sqrt{\hat{v}+\epsilon }},$$
where $\beta_{1}$ = 0.9, $\beta_{2}$ = 0.999, ϵ = 1 × 10^−8^, $v_{0}=0$, and $m_{0}=0$; $x_{i}$ is an orbit element, $g_{i}$ is the partial derivative of the RMSE with respect to element $x_{i}$, and t is the number of iterations.

### 2.2. Ballistic Coefficient Calculation from Mean Elements

According to Picone et al. [26,27], the ballistic coefficient can be calculated as
(9)$$\frac{da_{D}}{dt}=\frac{2a^{2}v}{\mu }\dot{v_{D}}\cdot e_{v},$$
where a is the semi-major axis, μ is the product of the gravitational constant and Earth mass, and $\dot{v_{D}}$ is the vector of the drag acceleration on space debris given by
(10)$$\dot{v_{D}}=-\frac{1}{2}\rho B{\left|v-V\right|}^{2}e_{v-V},$$

By integrating Equation (10) from time $t_{1}$ to $t_{2}$, we obtain
(11)$$∆a_{t1}^{t2}=\int_{t_{1}}^{t_{2}}Ba^{2}v\rho {\left|v-V\right|}^{2}e_{v-V}\cdot e_{v}dt,$$
where $∆a_{t1}^{t2}$ is the average semi-major-axis change caused by atmospheric drag from time $t_{1}$ to $t_{2}$. The corresponding numerical integration can be expressed as
(12)$$∆a_{t_{1}}^{t_{2}}=B\sum_{t_{1}}^{t_{2}}a_{t}^{2}v_{t}\rho_{t}{\left|v_{t}-V_{t}\right|}^{2}\cdot e_{v_{t}-V_{t}}\cdot e_{v_{t}}\cdot ∆t,$$

Therefore, the average ballistic coefficient from time $t_{1}$ to $t_{2}$ is given by
(13)$$B=\frac{a_{t_{2}}-a_{t_{1}}}{\sum_{t_{1}}^{t_{2}}a_{t}^{2}v_{t}\rho_{t}{\left|v_{t}-V_{t}\right|}^{2}\cdot e_{v_{t}-V_{t}}\cdot e_{v_{t}}\cdot ∆t},$$
where $a_{t_{2}}$ and $a_{t_{1}}$ can be determined as described in Section 2.1. The position and velocity of the target space debris at time t can be determined and calculated based on the orbit determination result from optical observations.

The atmospheric density of the target location at time t can be obtained by averaging multiple different atmospheric density models. In this study, we obtained the atmospheric density by averaging the results of three atmospheric density models: NRLMSISE00, DTM2000, and JB2006. Parameters $e_{v_{t}-V_{t}}$ and $v_{t}-V_{t}$ were calculated from the orbit determination results and atmospheric wind field model. We adopted the HWM96 wind field model. The instantaneous semi-major axis was calculated from the velocity and position parameters of the target space debris at each time as follows [28,29,30]:(14)$$a=r(2-\frac{rv^{2}}{\mu }{)}^{-1},$$
where r is the oblique distance of space debris and v is the debris velocity.

Overall, the ballistic coefficients of low-orbit space debris can be calculated using optical angle measurements from a photoelectric telescope array, as described in Figure 2.

## 3. Evaluation of Ballistic Coefficient Calculations

### 3.1. Verification of Mean Element Calculations

From optical observations acquired over 15–19 May 2021, we obtained the mean number of elements for NORAD TLE number 43,476. A comparison of the obtained prediction error and public TLE data prediction error is shown in Figure 3.

The dashed line in Figure 3 represents the prediction error of the mean elements of the target satellite at the corresponding time obtained by using the proposed method. The squares in the solid line represent the prediction errors of public NORAD TLE data, and the circles in the dashed line represent the extrapolation errors of the mean elements obtained from the experimental method. The NORAD number of the target space debris in Figure 3 is 43,476, as obtained from a satellite with laser ranging. The orbit calculation was conducted using laser ranging data to obtain the standard orbit. The NORAD TLE data were used in the SGP4 model to obtain the position parameters of the target, and the difference between these parameters and those of the standard orbit was determined to obtain the inversion error of TLE data at each time. The mean elements obtained from optical observations were also used in the SGP4 model to obtain the position parameters of the target over a specific period, and the inversion error at each time in that period was obtained through comparisons with the standard orbit. The public NORAD TLE data used for comparison are as follows in Table 1.

The experimental calculation results in TLE format are as follows in Table 2.

The error curve obtained using the proposed method is shown in Figure 3. The origin of the x-axis indicates the time of the mean elements, when the position error of the public data was 1046.24 m, while that of the experimental calculation was 584.51 m, representing an error reduction of 40.08%. Hence, the mean elements obtained using the proposed method were more accurate than the published values at the target time. Considering previous methods for calculating ballistic coefficients using public NORAD TLE data [2], the accuracy of the mean elements obtained by combining optical angle measurements with gradient descent for orbit determination is suitable for calculating the ballistic coefficients.

### 3.2. Verification of Ballistic Coefficient Calculations

To verify the effectiveness of the proposed method for ballistic coefficient calculation, the orbit prediction method was used for comparison. The following procedure was adopted:(1)The publicly available TLE data over the study period were obtained to be verified.(2)The public SGP4 model was used to calculate the TLEs of two-line elements and set a period to predict the ephemeris of public data.(3)The ballistic coefficient of public TLEs during the experiment period was replaced by the ballistic coefficient of the space debris obtained from the experimental calculation. The parameters contained in the TLE data, such as six orbital mean elements, remained unchanged, thus enabling us to obtain new TLE data to verify the calculation results.(4)The public SGP4 model was used to calculate the TLEs of the experimental calculation results and set a period to predict the ephemeris of experimental calculations.(5)The space debris statistical orbit determination result obtained from optical angle measurements was used as the true value of orbit parameters. The RMSE was calculated using the predicted ephemeris and true orbit value, to evaluate the quality of the experimental results. The RMSE is given by
(15)$$RMS=\sqrt{\frac{\sum_{t_{0}}^{t_{n}}{\left(X_{O}-X_{C}\right)}^{2}+{\left(Y_{O}-Y_{C}\right)}^{2}+{\left(Z_{O}-Z_{C}\right)}^{2}}{n}},$$
where C is the position parameter of the extrapolated time and O is the position parameter of the orbit determination result.

The detection accuracy of the equipment mentioned in Section 2 is 7.65 root mean square error. The satellite laser ranging data publicly released by the ILRS were used for orbit determination calculation, and the orbit determination results were used as the standard orbit. The accuracy evaluation of the orbit determination results of the optical measurements using standard orbits is shown in Figure 4 as follows.

The root mean square error of optical measurement orbit determination for target 46,469 (HY2C) in a time span of 1000 min is 59.02 m.

Different RMSE values were obtained from different ballistic coefficients. The verification results are shown in Figure 5, Figure 6, Figure 7 and Figure 8.

In Figure 5, Figure 6, Figure 7 and Figure 8, the squares in the solid lines represent the ephemeris error predicted from the target ballistic coefficient of public NORAD TLE data, while the circles in the dashed lines represent the ephemeris error predicted from the ballistic coefficient of TLE data calculated using the proposed method. An interval of 50 min was considered between datapoints. The RMSE of different space debris between the published ballistic coefficients and experimental results are listed in Table 3. The table shows the results on the same space debris over 24 h of prediction.

Figure 9 and Figure 10 show the error of orbit prediction using the public TLE data and ballistic coefficient (solid lines with squares) and that with the ballistic coefficient calculated by our method (dashed line with circles).

The 48 h prediction RMSE of NORAD space debris 25,876 using the published values was 1402.82 m for a ballistic coefficient of 0.00016351. The ballistic coefficient obtained from our experimental calculation was 0.00084, giving a 48 h prediction RMSE of 1259.98 m, which was 10.18% lower than the published error. The 48 h prediction RMSE of NORAD space debris 48,317 using the published values was 41,946.01 m for a ballistic coefficient of −0.0062747. The ballistic coefficient obtained from our experimental calculation was 0.008298, and the RMSE was 34,121.36 m, which was 18.65% lower than the published error.

## 4. Discussion

As shown in Figure 3 and Section 3.1, when using the proposed method described in Section 2.1, the mean elements obtained were more accurate than the published values at the target time, representing an error reduction of 40.08%.

As shown in Table 3 and Figure 5, Figure 6, Figure 7 and Figure 8, when using the ballistic coefficients calculated with the proposed experimental method for orbit prediction, the 24 h prediction RMSE was smaller than that obtained using the published ballistic coefficients. Although the reduction was not significant, it was still proven that this method could achieve an accuracy no less than that of publicly available data. As shown in Figure 8 and Figure 9, when using the ballistic coefficients calculated using the proposed experimental method for orbit prediction, the 48 h prediction RMSE was smaller than that obtained using the published ballistic coefficients. These results indicate the effectiveness of the proposed method for ballistic coefficient calculation.

## 5. Conclusions

To improve the accuracy when calculating the ballistic coefficients of space debris, we propose a calculation method based on optical angle measurements. The proposed method uses optical angle measurements to determine the position and velocity parameters of space debris during a specific period through precise orbit determination and provides the ballistic coefficients of space debris based on the calculated parameters. According to the verification of the ballistic coefficient calculation results of NORAD space debris 25,876, 39,093, 40,342, and 44,722, the calculated ballistic coefficients of space debris provide a reduction of 18.65% on the predicted ephemeris RMSE when compared with the published values for 15 May 2021. After verification, the ballistic coefficients obtained from our method were found to be more accurate than the publicly available ones, demonstrating the effectiveness and applicability of the proposed method. The next research direction is to investigate the relationship between the accuracy of ballistic coefficient calculation results and the accuracy of an optical observation system, in order to further improve the calculation accuracy.

## Figures and Tables

**Figure 1 sensors-23-07668-f001:**
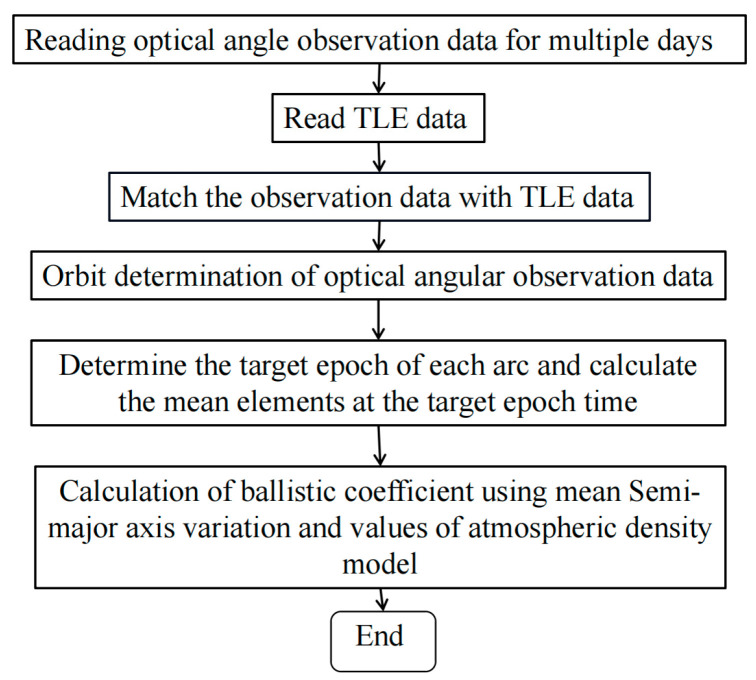
Flowchart for calculating mean elements of space debris from optical angle measurements.

**Figure 2 sensors-23-07668-f002:**
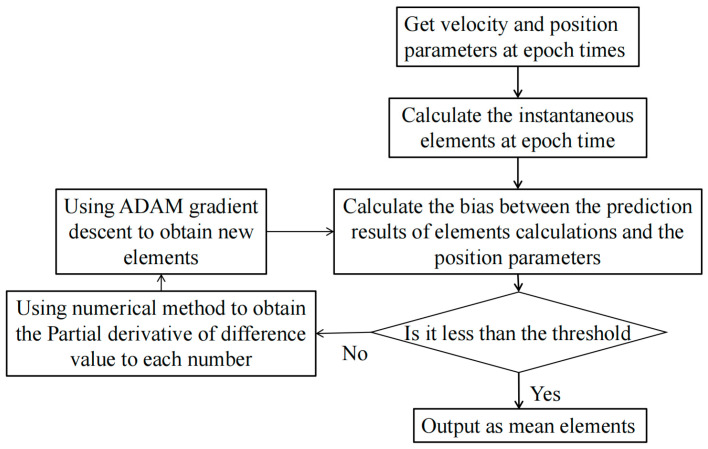
Flowchart for calculating ballistic coefficients of space debris.

**Figure 3 sensors-23-07668-f003:**
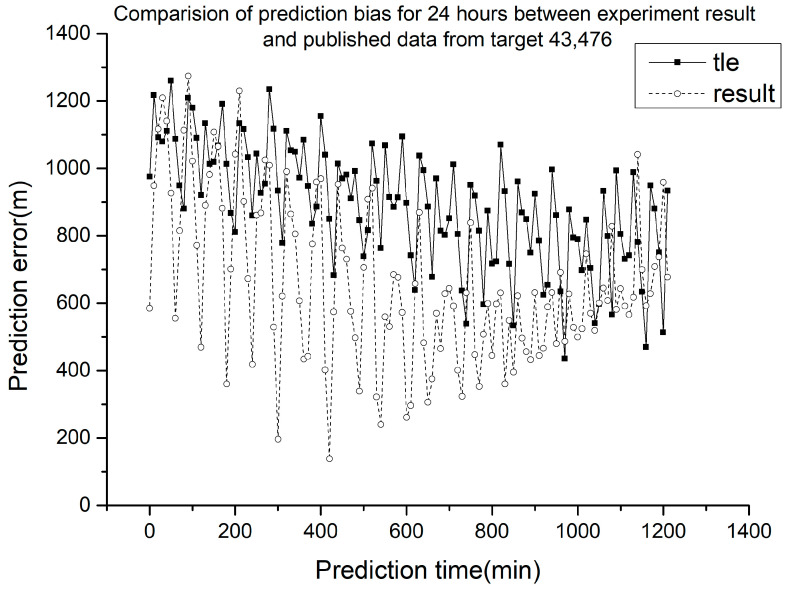
Prediction error of mean elements.

**Figure 4 sensors-23-07668-f004:**
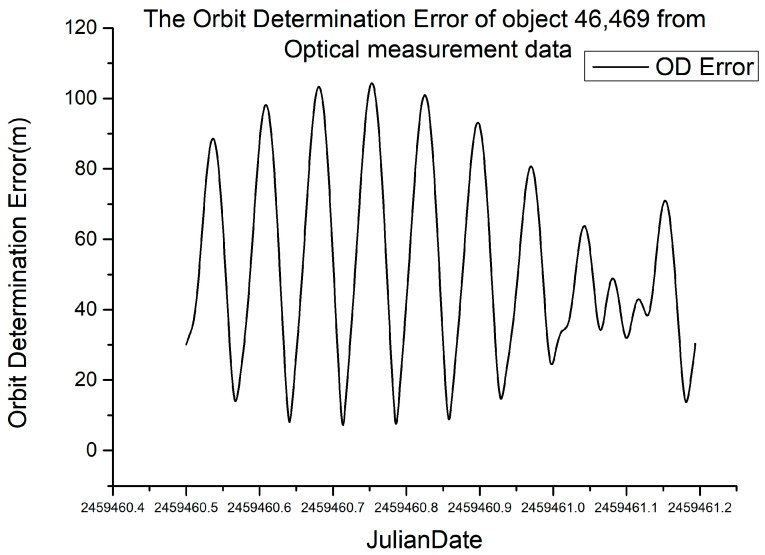
Orbit determination error of optical measurements for target 46,469.

**Figure 5 sensors-23-07668-f005:**
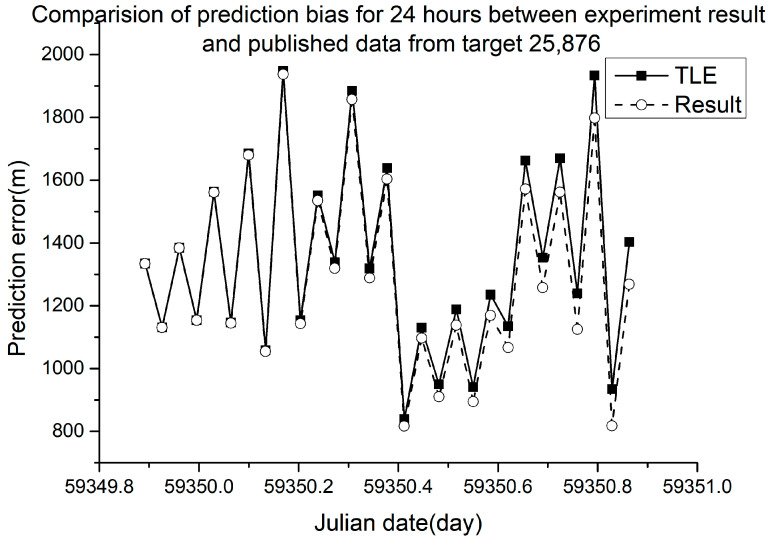
Orbit prediction error from debris 25,876 with different ballistic coefficients.

**Figure 6 sensors-23-07668-f006:**
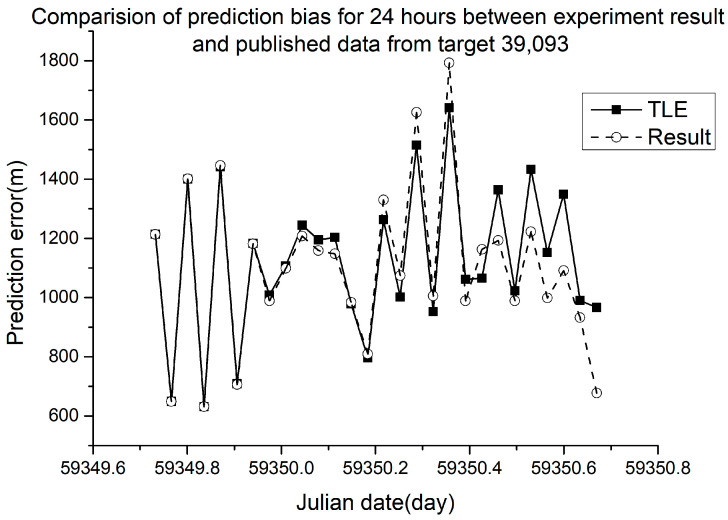
Orbit prediction error from debris 39,093 with different ballistic coefficients.

**Figure 7 sensors-23-07668-f007:**
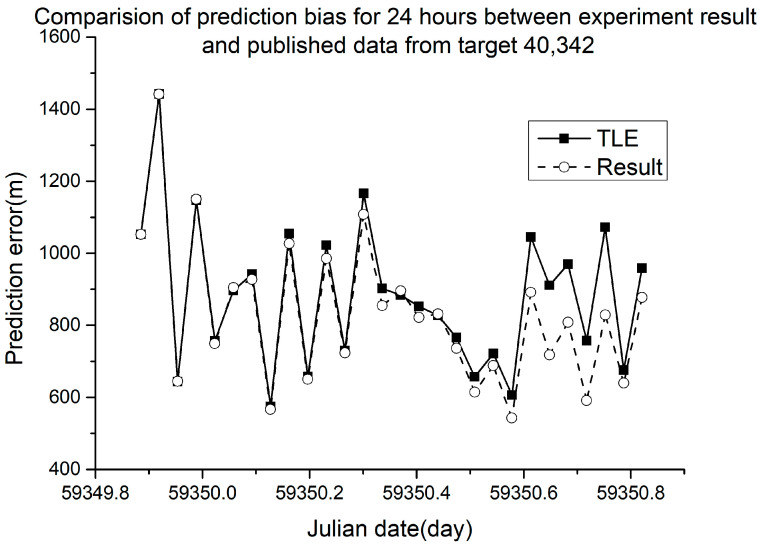
Orbit prediction error from debris 40,342 with different ballistic coefficients.

**Figure 8 sensors-23-07668-f008:**
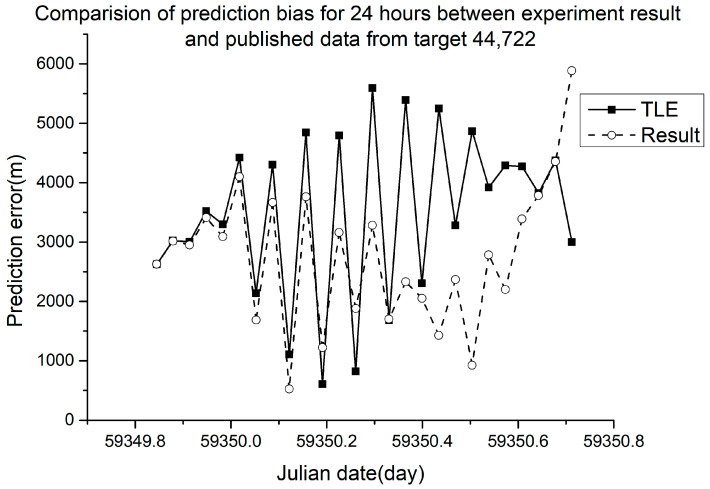
Orbit prediction error from debris 44,722 with different ballistic coefficients.

**Figure 9 sensors-23-07668-f009:**
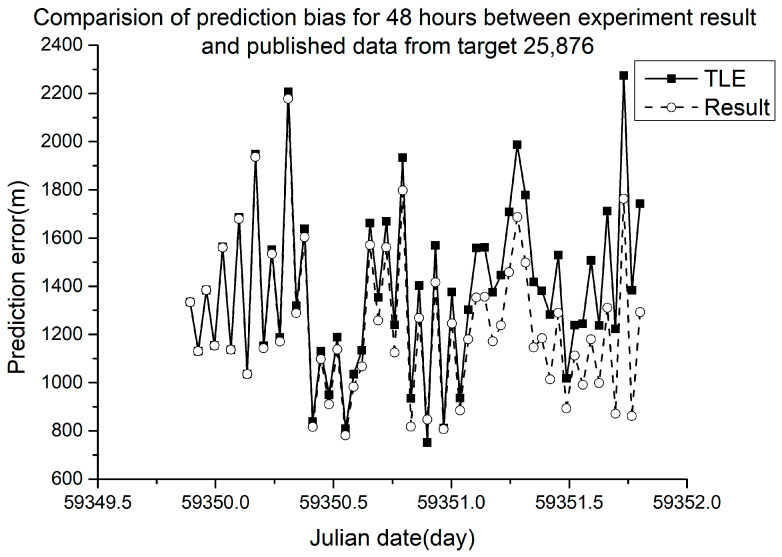
Orbit prediction error for 48 h from debris 25,876 with different ballistic coefficients.

**Figure 10 sensors-23-07668-f010:**
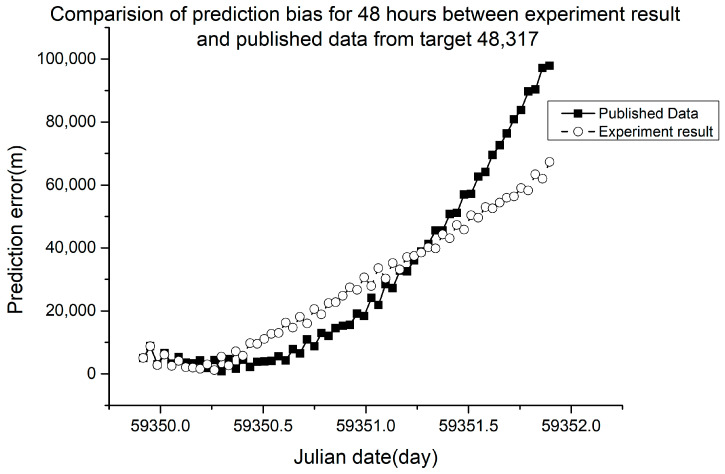
Orbit prediction error for 48 h from debris 48,317 with different ballistic coefficients.

**Table 1 sensors-23-07668-t001:** The public NORAD TLE data.

GRACE-FO 1
1 43476U 18047A 21135.59558637 0.00000467 00000-0 17718-4 0 9999
2 43476 88.9800 92.6669 0020229 75.3981 284.9508 15.24255690165800

**Table 2 sensors-23-07668-t002:** The experimental calculation TLE data.

GRACE-FO 1
1 43476U 18047A 21138.50000000 0.00000467 00000-0 15994-2 0 9999
2 43476 88.9767 92.2695 0018860 65.8255 020.8684 15.24261990165800

**Table 3 sensors-23-07668-t003:** RMSE of 24 h orbit prediction for space debris with different ballistic coefficients.

NORAD ID	RMSE from Published Ballistic Coefficient (m)	RMSE from Experimental Result (m)	Error Reduction (%)
25,876	1384.65	1342.22	3.06
39,093	1124.99	1114.06	0.97
40,342	917.98	904.05	1.52
44,722	3773.58	3652.85	3.20

## Data Availability

Not applicable.

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
