# Peer review of "Ballistic Coefficient Calculation Based on Optical Angle Measurements of Space Debris"

_sensors, 2023, doi:10.3390/s23187668_

Round 1

Reviewer 1 Report

11.       The study propose a method for calculating the ballistic coefficients of space debris using optical angle measurements. According to study the data acquisition of this method is more convenient compared to radar measurement data

22.       The topic is original and  relevant in the field of journal.   

33.       The study add to area;the average ballistic coefficient of the corresponding space debris arc segment is calculate using the change of the semi-major axis of the corresponding elements and atmospheric model.

44.       What specific improvements should the authors consider regarding the methodology? What further controls should be considered?

-Please add methodology section after introduction  to manuscript

-What is simplified general perturbation model (SGP4) please explain  at methodology section

- Please add a figüre for coordinate system  for line 68 (?1 , ?2 , ?3 , ?4 , ?5 , ?6) = ?(?1 , ?2 , ?3 , ?4 , ?5 , ?6),

- What is a,e, i, Ω, ω, M. Please explain these expressions ( line 70)

5. Please add more  references to introduction section.

Author Response

Thank you very much for suggestion.The article has been revised according to your suggestion.Please see the attachment.

Reviewer 2 Report

The paper presents a method to estimate the ballistic coefficient of space debris based on optical angular measurements.

General comments: The paper doesn't provide sufficient information to correctly estimate the quality of the work. The comparison with the actual TLE method is not complete, since, for example, there are no information regarding the assumpion on the errors on the optical measuraments required or performed. The quality of the description shall be improved.

Specific comments:

Throught all the paper: 

1) I sugget to replace "angle measuraments" with "angular measuraments".

2) I suggest to remove the 'dot' symbol in all the equations 

3) I suggest to remove the thousands separator (17,000 -> 17000)

4) I suggest to improve the editing quality of the equations

5) I suggest to improve the quality of the english language and on the terminology 

Title: The title comntains two errors: angel -> angle; Mearuements -> Measurements 

Line 23: 'general space targets' -> 'generic body'

Line 24: damping -> drag. windward .... -> cross sectional area with respect to the velocity direction

Line 26: space target -> object. The ballistic coefficient is also depending on the attitude motion of the spacecraft, not onl on its properties.

Line 47: typo follows.First -> follows. First 

Line 50: I suggest to add a reference for the Observatory mentioned.

Figure 1: remove watermark from image

Line 123: remove \

Figure 2: remove watermark from image

Figure 3: I suggest adding two lines that represent the filtered values for both TLE and Results, for example with a properly calibrated moving average filter

Line 176: remove \

The English launguage needs to be improved throught all the paper, as well as the terminology.

Author Response

Thank you very much for your suggestion.The article has been revised according to your suggestion.Please see the attachment.

Round 2

Reviewer 2 Report

The authors improved the papers following the suggestions.

There are some typos, for example:

line 45: a stpace after the "." is missing

line 49 and 51 58 59 88 167 194: idem.

Author Response

Thank you very much for your suggestion. Modifications have been made based on your suggestions. All the modifications have been highlighted. Please see the attachment.
